# AI Model for Predicting Legal Judgments to Improve Accuracy and Explainability of Online Privacy Invasion Cases

**Minjung Park** [ID] **and Sangmi Chai** *

Ewha School of Business, Ewha Womans University, 52 Ewhayeodae-gil, Seodaemun-gu, Seoul 03760, Korea; mjpark6767@ewha.ac.kr
* Correspondence: smchai@ewha.ac.kr; Tel.: +82-2-3277-2780

**Abstract:** Since there are growing concerns regarding online privacy, firms may have the risk of being involved in various privacy infringement cases resulting in legal causations. If firms are aware of consequences from possible cases of invasion of online privacy, they can more actively prevent future online privacy infringements. Thus, this study attempts to predict the probability of judgment types caused by various invasions within US judicial cases that are related to online privacy invasions. Since legal judgment results are significantly influenced by societal factors and technological development, this study tries to identify a model that can accurately predict legal judgment with explainability. To archive the study objective, it compares the prediction performance by applying five types of classification algorithms (LDA, NNET, CART, SVM, and random forest) of machine learning. We also examined the relationship between privacy infringement factors and adjudications by applying network text analysis. The results indicate that firms could have a high possibility of both civil and criminal law responsibilities if they distributed malware or spyware, intentionally or non-intentionally, to collect unauthorized data. It addresses the needs of reflecting both quantitative and qualitative approach for establishing automatic legal systems for improving its accuracy based on the socio-technical perspective.

**Keywords:** online privacy invasions; personal information infringements; predicting judgments; predictive analytics; privacy act; network text analysis

## 1. Introduction

Prediction of legal judgment is a long-lasting topic in the theory and practice of law to improve judicial consistency, access to justice, and administrative efficiency [1]. Therefore, various methods and techniques have emerged over time, including simple calculative models to highly advanced analytical algorithms to predict legal judgements. There have been a wide range of approaches attempted; in particular, artificial intelligence (AI)-based approaches have been increasingly utilized, with the recent advent of AI.

Legal decision systems have been established, based on AI, by predicting verdicts automatically to support lawyers.

A legal automation system called AI lawyer was invented to predict verdicts in the United States on May 2016, and since, constant efforts have been made to develop its accuracy. It reads a vast number of judgment documents and analyzes the contents based on a special algorithm to draw decisions to judge a case automatically. It becomes more difficult and time consuming for human lawyers to sentence correct verdicts in legal judgments, because there have been significantly increased numbers of lawsuits in the recent past. Thus, there is a growing need to develop systems for predicting legal judgments precisely based on a vast number of legal precedents, with the emergence of these AI lawyers and automatic legal systems based on big data. However, there is little research on big data analytics in the field of law. Therefore, our aim is to provide an AI model for establishing a system to predict legal judgments with explainability by identifying

factors with high potential to cause privacy infringements and thus constitute illegal acts, as well as to compare the performance of predicting judgments.

This study focuses on the judicial cases related to privacy infringements caused by firms [2]. Since online privacy infringement cases cause severe legal consequences for firms, it is important for them to be aware of various privacy infringement with legal liability. In addition, as the number of online privacy invasion cases is predicted to increase and become more diverse, the development of an automated predictive model for legal decisions can significantly reduce the efforts of legal practitioners as well as companies. Therefore, this study is devised from the motivation to suggest the foundation for applying and interpreting a predictive model from a socio-technical perspective, along with deriving an optimal judgment model.

Legal cases related to privacy infringement are outcomes of interactions between society and technological factors, since the cases often comprise violation of law, human errors, personal information owners' perception, etc. It means that recent privacy legal cases have shown different characteristics from other legal incidents. For example, the case of stealing someone's property is always considered illegal, regardless of time and space. However, in most privacy infringement cases, it is very difficult to clearly identify a responsible party. Legal cases regarding improper usage of adware to invade an individual's privacy is a representative case that demonstrates that legal judgments may be influenced by technological and social environmental factors. Therefore, this study applied NTA (network text analysis) method, which has been used in sociology to reflect social influences [3]. In addition, we performed a comparison of AI models, including LDA (linear discriminant analysis), NNET (neural network), CART (classification and regression tree), SVM (support vector machine), and random forest, to identify a model with high prediction accuracy. The results of this study provide the foundation for developing an automated legal prediction system that could consider influences from social and technical factors that past research did not consider. This study is conducted in the following steps. The discussion of an extensive literature review of judgement prediction and algorithms for online privacy is performed in Section 2. We introduce the collected data, the characteristics of our collected data, and the five classifications methods (LDA, NNET, CART, SVM, random forests) for predicting in Section 3, the research method section. In Section 4, the comparison of the performance of each model is presented. In Section 5, constructed networks of legal judgments by NTA are discussed. We suggest concluding remarks, as well as contributions and limitations of this study, respectively, in Sections 6 and 7.

## 2. Related Works

### 2.1. Prediction of Legal Judgments

The advancement in natural language processing and machine learning is contributing to predictive models that identify various patterns in judicial decisions. Related studies in predicting legal judgments mainly take two approaches: increasing the accuracy of predictions under the present algorithms or predicting the outcome of future legal disputes by using statistics and AI in real judicial cases.

A prediction model based on contiguous word sequences (i.e., N-grams and topics) has up to a 79% rate of accuracy and is now applied to cases in the European Court of Human Rights [4]. Prior research mainly focused on textual information. However, IBP (issue-based prediction) has recently been used to predict legal outcomes. Although IBP is similar to previous computer models, in that it predicts legal judgments based on statistics and AI, it has an ability to draw an overall prediction through testing assumptions [5]. When IBP is applied to a legal case, it establishes favorable assumptions for each participant in the case. To verify the assumption, IBP infers large numbers of past legal judgments [6]. A new model, CNN-BiGRU, is suggested to have better prediction than a single CNN or RNN model [7], and we propose the generalized Gini-PLS algorithm, which is based on the simple Gini-PLS model, to develop a judicial prediction system [8].

### 2.2. Algorithms for Protecting Online Privacy

Research on online privacy has focused on personal information that can be leaked in an e-commerce environment and online social networks [9]. These days, a massive amount of personal information has been collected and managed online, and it has raised privacy concerns about how to manage these data appropriately [10]. Fast-developing IT technology also causes various types of privacy invasions, which could further increase related incidents. Online privacy studies in the field of engineering provide technical measures and prediction algorithms against privacy invasion. A study by Hanguang and Yu [11] developed a web-based intrusion detection system to identify external attacks and improve the overall performance of the detection system through an algorithm, Apriori. One study designed a system to detect personal information stored in a user's PC and evaluated the impact based on the possibility of data breach [12]. The research of Blei and Ng [13] explored topic modeling techniques to develop an algorithm for a privacy invasion forecast system. Further, as there are growing concerns about re-identification of personal data online, related research is also being actively conducted [14].

A lot of research has been conducted to determine a method to protect online privacy by developing protection techniques such as differential privacy (DP), which has been introduced to preserve privacy in datasets [15,16]. It is defined as a way of circumventing the problems of an adversary with auxiliary information and provides the level of privacy with superior performance [10]. DP was based on a probability model with a set of conditions that need to be met to guarantee that auxiliary information will not result in a privacy breach [17,18]. It has the ability to create useful statistics by itself, while the users' privacy is maintained in a database [19].

A new algorithm has recently emerged for lowering error by adapting to properties of the input data, so-called data-dependent algorithms [16]. Pythia proposes a meta-algorithm that does not need to understand valid algorithms or identify the subtle properties of input data [10]. Functional dependency (FD) is based on preserving probabilistic encryption scheme. It considers the frequency analysis (FA) attack and the functional dependency preserving chosen plaintext attack (FCPA) for protecting sensitive information in the outsourced data while preserving the data dependency for the data owner [20].

K-anonymity, K-privacy, and K-support techniques have been applied to data to protect the privacy of the outsourced database and the mined the association rule [17]. However, these techniques have a shortcoming, in that they are relatively expensive. To overcome this drawback, Yi and Rao [21] proposed a solution by performing association rule mining on the encrypted data in the cloud and returning encrypted association rules to the user in the same time. The data-cleaning-as-a-service (DCaS) paradigm makes users outsource their data and attain a data cleaning service by third-party service providers [20]. However, this paradigm raises the issue that it cannot be guaranteed that their private information in the outsourced data is fully protected. To solve this problem, Dong and Liu [20] designed the privacy-preserving data-deduplication-as-a-service (PraDa) system, focused on data deduplication, which is the most vulnerable in data cleaning problems. By providing privacy assurance against both the known-scheme attacks and frequency analysis, PraDa secures the server to find duplicated records from the encoded data [20].

Previous studies have predicted judicial decisions by focusing only on textual information without considering the social environment. That is, the adjudication is predicted by a fixed the pattern between words and decisions, which is constructed based on a specific word frequency. It is also made without including the characteristics of various online privacy intrusion factors. To overcome the limitations of prior studies, this study attempts to predict the judgments in a social context, focusing on each factor of online privacy invasion, and finally, to provide the explainability of our prediction models. To this end, we intend to establish a social network of the judgment and each factor of online privacy intrusion and provide a foundation for explaining how each factor affects the judgment.

## 3. Research Method

### 3.1. Data Preparation

The Privacy Act in the United States takes different approaches to the public sector and the private, unlike other regions such as Europe. This sectoral approach has the advantage that the law can promptly respond to new social issues and IT. Online privacy invasion factors vary by environment, so this study selected the legal precedents of the United States, which adopted a sectoral approach as the target of analysis. Therefore, this study analyzed United States judicial cases, which were collected from Westlaw database. Westlaw is a database that contains legal documentary data in the U.S. It classifies data based on key issues of judgments [22]. Among various classification items, this study collected 1098 cases of legal precedents (from January 2000 to December 2018) to obtain judgment documents related to privacy invasions.

We collected data from Westlaw, which include only federal law precedents among numerous online privacy invasions across the United States. A federal law is applied to the nation as a whole and to all 50 states, whereas state laws are only in effect within that particular state. Therefore, in this study, only the precedents that have been sentenced based on federal law were selected for analysis, as that can be applied equally to all states in the United States. The reason that only cases sentenced by federal law are selected for this study is that it focuses on online privacy invasions that occurred in a virtual online environment without physical territorial limitations. In other words, this study deals with cases sentenced by federal law, regulating without physical territorial restrictions to reflect the features of online spaces rather than dealing with sentences under state law, including the regional characteristics in which the case occurred. Therefore, it is expected that the results of this study can be applied to online privacy infringement cases occurring across the United States without distinction of state law.

Table 1 shows the numbers of the legal precedents each year we collected from Westlaw. The highest number of sentences related to online privacy invasions were decided under federal law in 2012 with 87, followed by 83 in 2018.

**Table 1.** The Numbers of Precedents of Each Year.

| Year | Precedents | Year | Precedents | Year | Precedents | Year | Precedents |
|------|-----------|------|-----------|------|-----------|------|-----------|
| 2000 | 42 | 2005 | 41 | 2010 | 51 | 2015 | 51 |
| 2001 | 53 | 2006 | 54 | 2011 | 53 | 2016 | 56 |
| 2002 | 62 | 2007 | 44 | 2012 | 87 | 2017 | 77 |
| 2003 | 61 | 2008 | 74 | 2013 | 56 | 2018 | 83 |
| 2004 | 52 | 2009 | 49 | 2014 | 62 | | |

To examine the interconnectivity between the words extracted from text preprocessing and judgments related to privacy invasion, they were classified into two groups as follows. First, actual malicious codes generated to collect personal data from other PCs without the users' consents are classified as "type of privacy invasion". Second, words relevant to the actual adjudication of precedents are classified as "type of privacy invasion judgment". The authors classified those words into 2 categories according to the research's main objective. For example, "type of privacy invasion" mainly consists of techniques that hold possibilities of privacy threats. The case of "type of privacy invasion judgment" is composed of terminology (i.e., conviction, compensation, innocence) that is usually used in verdicts for presenting the results of litigations. Table 2 shows that words are finally extracted to construct networks and provide meaning.

**Table 2.** Definition and Frequency of Privacy Invasions.

| Type of Privacy Invasion | Definition and Characteristics | Frequency |
|---|---|---|
| Adware | Refers to software that randomly shows advertisements to users but can also be used to collect personal data [23] | 124 |
| Cyberattack | An action causing damage to the other party's company by invading the user's PC through the internet. It is used as a general term for illegal access behaviors [24] | 79 |
| | A type of cyber terrorism such as leaking personal data or crashing websites for political or social purposes [25] | |
| Malware | A program that causes failures in system operation, acquires unauthorized access to data collection or system resources, or is used for other acts of invasion [3] | 195 |
| Spam | Refers to for-profit advertising data that are sent in unsolicited bulk, without consent, to devices such as email or cell phones of users of information and communication services [26] | 212 |
| Spyware | Software that collects personal data after being installed without the user's consent by deceiving the user [27] | 137 |
| Vandalism | Acts of destroying order in cyberspace that threaten personal data by posting another person's data with misuse of anonymity or through defamation of a specific person [28] | 45 |
| Virus | An illegal program that destroys important data or software by invading the user's system or expanding damages through self-replication, using a network [29] | 184 |

Types of privacy invasion judgments include conviction and innocence representing adjudication in terms of criminal law, and imprisonment, penalty, and probation, which are criminal punishments. Imprisonment is a measure that restrains a person's freedom by confining the suspect or convict in a restricted space such as a detention center or prison; it indicates a prison sentence or confinement. Probation is a system that improves and rehabilitates the criminal under certain supervision and guidance with a free social life, without confining him or her in prison.

*3.2. Classification Techniques in Machine Learning*

Machine learning is a vast interdisciplinary field, which is based on concepts from statistics, computer science, cognitive science, engineering, optimization theory, and many other disciplines of mathematics and science. In machine learning, supervised learning algorithms—labelled training datasets—are used first to train the underlying algorithm. This trained algorithm is then provided on the unlabeled test dataset. Supervised learning algorithms that deal with classification include the following five representative algorithms: LDA (linear discriminant analysis), neural networks (NNET), classification and regression tree (CART), SVM (support vector machine), and random forests. This is used for classification analysis through a learning algorithm that makes predictions for unexperienced or future data. Classification predictive modeling in machine learning is an approach used to predict binary data. It is an approach that is used for purposes such as classifying emails into "spam" or "non-spam" to filter them automatically [30] or to predict whether it will rain or not [31]. Furthermore, classification predictive model has been widely used in various fields, for example, to predict whether to buy or not, whether or not customers belong to a group membership, or to classify images into two groups.

We select five state-of-the-art classification algorithms (i.e., LDA, NNET, CART, SVM, random forests) for which performance of binary classification has been verified on text data as well as numerical data among various models. The reason for adopting those

five algorithms is, first, as the primary purpose of this study is to compare the legal judgment prediction performance of various binary classification models, it is necessary to adopt models that have been repeatedly verified in relatively various fields. Secondly, considering our data set, models that were not suitable for best performance in our data set were excluded. Traditional binary classification models such as logistic regression or stochastic gradient descent were not adopted in this study, as they rarely show high predictive performance rate compared to the other binary classification models in a small dataset, although they have been verified and improved over a long period [12]. The main purpose of this study is to discover the best prediction performance for legal predictions, since the number of our data sets in terms of cases number itself are relatively smaller than other studies using AI algorithms; algorithms such as logistic regression or stochastic gradient descent were not suitable, as they cannot used for self-training data set [12]. Naive Bayes classifier is also not applied to this study, since Naive Bayes model is not appropriate for data with binary values such as 0 and 1 due to the zero frequency problem [32]. We finally adopt five representative classification algorithms in supervised learning as LDA, CART, NNET, SVM and random forests in this paper among various models for binary classification. In the following paragraphs, each classification algorithm adopted in this study will be introduced in order of the logical flow of each definition and model construction process. It also includes the process for performing classification prediction in accordance with the purpose of this study.

### 3.2.1. Linear Discriminant Analysis (LDA)

Fisher's linear discriminant analysis (LDA) constructs discriminant functions that estimate discriminant values for each of subjects classified into a certain group from linearly independent predictor variables [33]. These discriminant weights are calculated by ordinary least squares, so that the ratio of the variance within the $\lambda$ groups to the variance between the $\lambda$ groups is minimal [34]. The decision boundary defined by LDA is linear, and that defined by QDA is quadratic. LDA is limited in flexibility compared to the other classification methods when applied to more complex datasets [35]. LDA generally shows similar performance with logistic regression [36].

### 3.2.2. Classification and Regression Tree (CART)

Classification and regression trees (CART) construct hierarchical decision trees by splitting data among classes of the criterion at a given node accordingly to an "if-then", applied to a set of predictors, into two child nodes repeatedly [37]. Classification trees are verified as an appropriate method for predicting the binary of target variables with high accuracy and require few assumptions about the data [38,39]. It keeps partitioning data with explanatory variables in binary split, which gives the minimal impurity until the terminal node has a predefined minimum size [40,41]. Then, it fits the response variable in each partition. The goal is to find a partition, so that the response is the most homogeneous in each partition [42]. A CART model that predicts continuous variables from a continuous or categorical predictor variable is referred to as regression model [43,44]. Decision tree-based models, including CART, have an advantage in that they are scalable to large problems and can handle smaller data sets than NN models [41]. The more complex model has the better prediction power; however, a too complex model can be hard to be interpreted, and there can be an overfitting problem [16]. Therefore, the most important aspect of constructing CART is to have balance between complexity and goodness of fit.

### 3.2.3. Neural Networks (NNET)

Neural networks have been applied extensively in both regression and classification problem. A neural network holds layers of interconnected nodes. Each node is a perceptron and is similar to a multiple linear regression. The perceptron feeds the signal produced by a multiple linear regression into an activation function, which is nonlinear [33]. It assumes that the response variable (output layer) has a relationship with explanatory variables

(input layer) and that there is hidden layer between them in a model [16]. It postulates that input layer affects all nodes in the hidden layer, and the response is affected by all nodes in hidden layer [34]. Neural networks are differentiated from existing optimization algorithms, in that they are identified in parallel by a group of vectors, and it does not depend excessively on the initial parameter but changes stochastically [35].

### 3.2.4. Support Vector Machines (SVM)

Support vector machines (SVM) find a linear separating hyperplane constructed from a vector x of predictors mapped into a higher dimension feature space by a nonlinear feature function [36,37,44]. It derives classifiers, which map a vector of predictors into a higher dimensional plane through either linear or non-linear kernel functions [45,46]. SVM shows fairly good prediction accuracy based on its sound theoretical foundation in complex classification problems [36]. It is designed to perform learning in the direction of minimizing 'structural risk'; on the other hand, ANN pursues 'empirical risk management'. SVM also requires a small amount of training data called support vector, generally using only a small amount of data and being insensitive to the number of dimensions for final learning, unlike ANN [36,44]. Therefore, SVM can be relatively free from overfitting problems and offers one of the most robust and accurate methods.

### 3.2.5. Random Forests

Random forests are similar to a decision tree or a bagging classifier, having the same hyper-parameters [47]. Therefore, it can be explained as a way of classification by adding an additional layer of randomness to bagging [48,49]. It constructs a series of CART using random bootstrap samples of the original data sample [37,48,49]. Each tree is formed from a random subset of the total predictors who maximize the classification criteria at each node. A classification error rate can be calculated using each of the CART to predict the data not in the bootstrap sample used to grow the tree, and then mean values as out-of-the bag predictions for the grown set of trees become a "forest". Random forests methods can be easily optimized by adjusting only two parameters. It requires defining the number of random trees in the forest and the number of predictor variables in the random subset of tree at each node [35]. Random forests can be used both for two-class and multi-class problems of more than two classes, and when there are many more variables than observations [50]. It has good predictive performance, even when most predictive variables are noise [50,51]. However, random forests have a possibility to detect over-detection of real and false paradoxes in subsets of the data that do not occur in the entire dataset due to sampling error [52].

### 3.3. Network Text Analysis (NTA)

Network analysis is a research technique that focuses on seeking the relationship patterns between agents and their related data [53,54]. Network text analysis (NTA) is a method that creates a model of nodes and links and quantitatively analyzes their phase structure as well as the process of diffusion and evolution [55]. Therefore, the network structure established through NTA is useful for interpreting certain social phenomena beyond the dictionary meaning of the relevant text [3]. According to Bhat and Milne (2008), degree centrality analysis is the most used to understand the influence of entities and structures in a network [56]. The highest node has the strongest influence on spreading infectious diseases compared to the other nodes in the network [57]. NTA complexly connected the social network analysis technique with text analysis, and thus it simultaneously explores both knowledge and understanding of a given social phenomenon [9].

Legal verdicts inevitably reflect social phenomena (e.g., nations' characteristics, ethics, social values); thus, this study adopted NTA for analyzing precedents. More specifically, this study conducted keyword-based network analysis for in-depth analysis of legal precedents by extracting specific keywords presented in privacy legal cases, which can be considered a text aggregate of socio-structural phenomena, using NetMiner 4.0. Specifically,

a concept in the network consists of one or more related words, which are nodes in a social network analysis, while links represent the relationship between two concepts. In other words, two or more concepts simultaneously appearing in a single sentence indicates that the two have a close relationship, and the association of these linkages enables implementation of a semantic map [26]. That is, co-word analysis enables calculation of co-occurrence frequency of each word, as well as classification of the subject area in the relevant field, elicitation of keywords, and determination of correlation among subject areas.

## 4. Performance Comparison of Machine Learning

### 4.1. Classification Model Construction

We adopt five classification methods by R programs for predicting the judicial decision as introduced in Section 3. All these techniques are appropriate for predicting results of binary dependent variables (coded 0: innocence, 1: conviction). To find optimized hyperparameters of each machine learning model, its value is tuned as follows. First, the hyperparameter used for the optimal SVM model is combined with the penalty parameter (C) for 0.1, kernel coefficient ($\gamma$) of 0.03, and Kernel linear function. In random forests, the best performance is identified when the maximum number of features considered for splitting a node set is 50, and the maximum levels of each decision tree are set as 3. Weight decay was tuned between 0 and 0.1 and found the optimal value as 0.09 in size 18 in a model of NNET. In CART, the best tree size is determined to be 6 in the minimum deviance, as shown in Figure 1. We complied this with a basic LDA model, as it did not require any additional tuning.

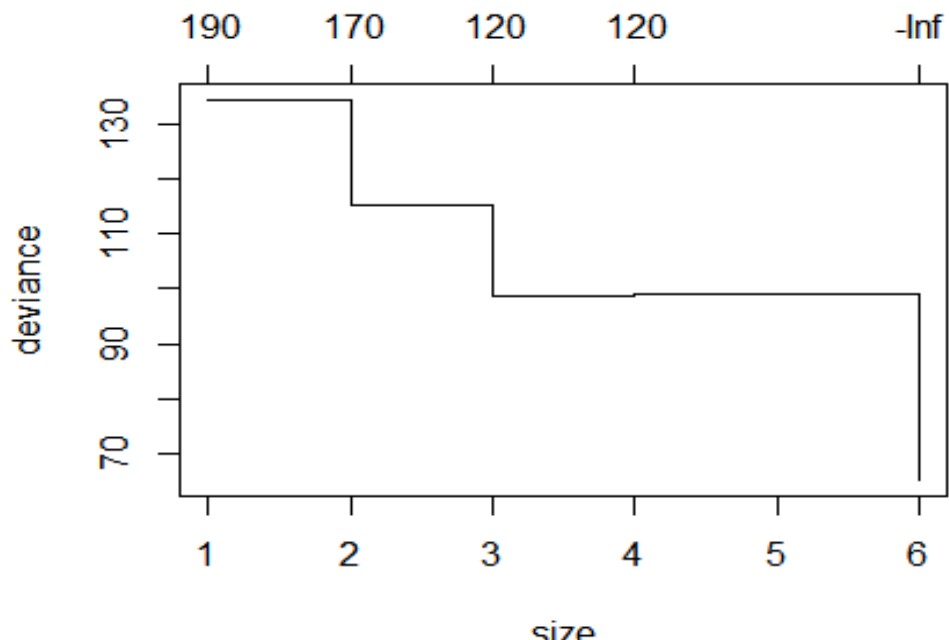

**Figure 1.** Deviance vs. Tree Size.

Table 3 provides the classification performance by each method as performed once. It shows the predictability, how each model can classify "real guilty case is judged as a conviction" and "real innocent case is judged as a guiltlessness". As a result, CART has the greatest prediction rate, while neural network performs worse than the others.

**Table 3.** Classification for Judicial Decisions.

|  | Classification | Innocence | Conviction | Prediction Rate (%) |
|---|---|---|---|---|
| LDA | Innocence | 451 | 85 | 82.33 |
|  | Conviction | 109 | 453 |  |
| CART | Innocence | 460 | 76 | 83.16 |
|  | Conviction | 109 | 453 |  |
| Neural Network | Innocence | 466 | 70 | 81.79 |
|  | Conviction | 130 | 432 |  |
| SVM | Innocence | 448 | 102 | 82.69 |
|  | Conviction | 88 | 460 |  |
| Random Forest | Innocence | 463 | 73 | 83.06 |
|  | Conviction | 113 | 449 |  |

### 4.2. Model Validation and Performance Comparison

Using the above optimized hyperparameters, each model can be trained and established. To obtain independent test data and reliable results, each data set was split randomly (70:30) into a training and a test data set. In other words, we use 70% of data for training as model fitting and 30% for evaluating, among the 1098 cases we collected. The training data were evaluated and validated through K-fold cross validation. To reduce the influence of a single sampling method on model results, the seven-fold cross-validation method was adopted to select training data and test data. Seven-fold cross-validation method divided the whole dataset into seven disjoint subsets randomly and averagely. Seven subsets was identified as an optimal value in this model, among the most popular and common value (e.g., k = 3, 5, 7, 10) used in applied machine learning [33]. To compare the classification models with judicial cases, we performed 100 simulations (i.e., experiments are repeated 100 times) of each model to generalize its accuracy and compute the mean of misclassification rates, as shown in Table 4 and Figure 2. The greatest accuracy of classifications was achieved by CART, showing the lowest misclassification rate to be as same as that analyzed simultaneously. As a result, CART has the best performance rate for predicting judicial decisions.

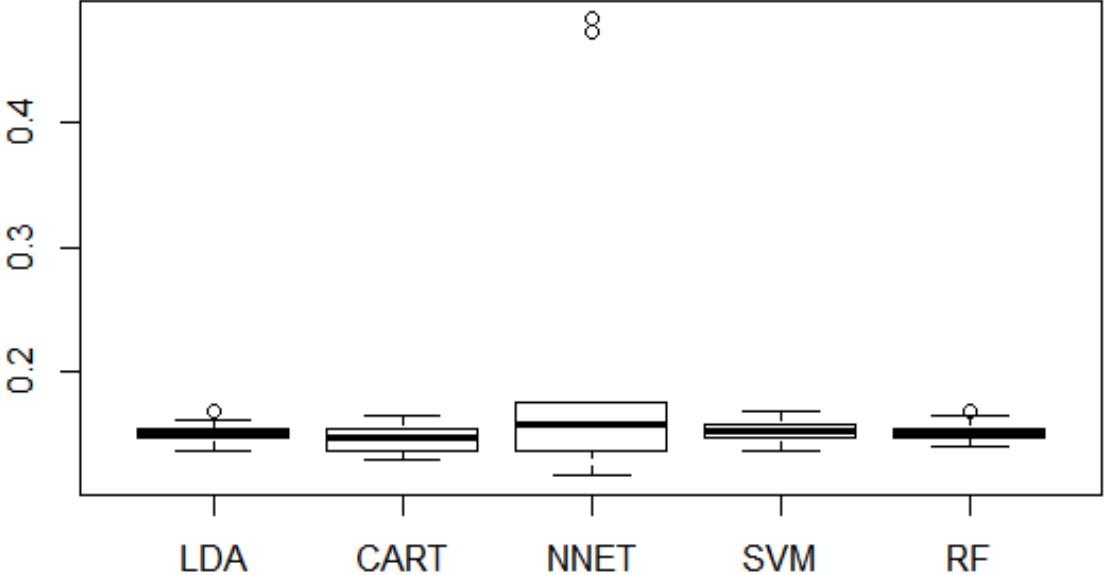

**Figure 2.** Misclassification Rates in 100 Simulations.

**Table 4.** Misclassification Rates in 100 Simulations.

| Methods | Misclassification Rate (%) |
|---|---|
| LDA | 11.82 |
| CART | 11.52 |
| Neural Network | 12.04 |
| Support Vector Machine | 11.90 |
| Random Forest | 11.72 |

## 5. Constructing Networks of Legal Judgments

Judicial cases refer to an aggregate of documents drawn up based on related regulations, using legal terms and focusing on specific cases or basic facts about the participants. Therefore, those forming a legal precedent are bound to have a close correlation among them. To analyze the relationship among words that form a legal precedent, this study conducted NTA. It is a descriptive modeling method for explaining the social phenomena in a macro perspective [3]. NTA is useful way to explain social issues by analyzing factors in social phenomenon [3], and it enables a researcher to identify the relationship among various confounding factors such as the factors invading privacy and the adjudication types. First, we formed a weight matrix based on co-occurrence frequency in a single paragraph as the baseline data for network analysis. Based on the elicited weight, this study also calculated the degree of centrality and concentric of the main keywords. Keywords with a greater value of degree centrality have greater influence in the overall network. This degree centrality is elicited based on an eigenvector; thus, there is a limitation, in that the frequency of the relevant keyword in the overall text is not considered. Accordingly, an analysis of concentric was conducted, additionally focused on the frequency of relevant keywords in legal precedents.

### 5.1. Analysis of the Possibility of Judgment According to Types of Privacy Invasion

To analyze the interrelation between types of privacy invasion and specific judgments, additional research was conducted by dividing the precedents into the possibility of conviction and innocence from the perspective of criminal law and possibility of compensation in terms of civil law.

#### 5.1.1. Possibility of Conviction and Innocence in Terms of Criminal Law

To form a network of judgment on conviction and innocence according to the types of privacy invasion in terms of criminal law, conviction and innocence were set as the central nodes, and the relationship between the two was examined. To form a network, the degree centrality of types of privacy invasion were calculated according to conviction and innocent, which are presented in Tables 5 and 6 below.

**Table 5.** Degree Centrality of Conviction.

| Main Keyword | Degree Centrality | Network of Privacy Invasion with Conviction |
|---|---|---|
| Adware | 0.155 | |
| Cyberattack | 0.018 | |
| Malware | 0.278 | |
| Spam | 0.266 | |
| Spyware | 0.230 | |
| Vandalism | 0.034 | |
| Virus | 0.191 | |

**Table 6.** Degree Centrality of Innocence.

| Main Keyword | Degree Centrality | Network of Privacy Invasion with Innocence |
|:---:|:---:|:---:|
| Adware | 0.143 | |
| Cyberattack | 0.085 | |
| Malware | 0.221 | |
| Spam | 0.142 | |
| Spyware | 0.429 | |
| Vandalism | 0.222 | |
| Virus | 0.102 | |

If a certain type of invasion forms a strong connection with the node of conviction through the formed network, a conviction is more likely to occur. The types of invasion that are marked with big circles or nodes in the network and are directly linked to adjudications such as conviction or innocence are likely to receive relevant verdicts. In other words, if nodes are relatively big, such as malware, and are directly connected in primary links with conviction in the network, the invasion has a high possibility to be judged illegal. Furthermore, it can be assumed that there is a slight chance of innocence, because the types of privacy invasion with the possibility of being convicted (e.g., virus, adware, spam, cyberattack, and malware) do not form a relationship with nodes focused on innocence.

5.1.2. Possibility of Judgment of Civil Compensation by Invasion Type

Compensation generally refers to the return to conditions present before the damage was done, with the purpose of recovery and relief of violation of one's rights. It is confirmed that all of types of privacy invasion factors form a direct relationship with compensation, and especially malware, which form large nodes in the network, are likely to receive the verdict of compensation. Finally, it can be verified that even the same type of privacy invasion is likely to be subject to different judgments and types of measures depending on specific environmental factors such as current conditions. Table 7 shows the degree of centrality and network of the types of invasion that are likely to receive the verdict of compensation as a means of relief for privacy invasion.

**Table 7.** Degree Centrality of Compensation.

| Main Keyword | Degree Centrality | Network of Privacy Invasion with Compensation |
|:---:|:---:|:---:|
| Adware | 0.173 | |
| Cyberattack | 0.198 | |
| Malware | 0.231 | |
| Vandalism | 0.098 | |
| Virus | 0.102 | |
| Spam | 0.200 | |
| Spyware | 0.211 | |

## 6. Conclusions

This study performed classification methods of machine learning for predicting judicial decisions and analyzed the network. With machine learning models, it is usually

very hard to explain how each factor influences the establishment of a model. Thus, this study constructed the network of judicial cases according to the privacy invasion factors by NTA. We could identify which model had the highest performance rate of prediction of legal decisions among the five classification algorithms and infer how these algorithms can predict legal or illegal activities by analyzing the network that was established by NTA.

The results of this study can be summarized as follows. First, the CART technique had the best performance of classification compared to the other methods. It can be inferred that CART algorithms should be considered a priority among various prediction algorithms for establishing the system of an AI lawyer or an automatic judicial decision machine. In other words, as CART shows the best performance within the scope of online privacy cases used in this study, it is desirable to adopt CART to predict legal disputes and judgments related to online privacy. These results have the potential to be used not only in online privacy disputes but also in resolving various legal conflicts that may occur in the online environment. However, since there is no guarantee that CART always shows the highest performance rate in predicting the guilt or innocence of numerous off-line crimes or in predicting legal decisions (e.g., probation, penalty) not covered in this study, other models should also be considered. Secondly, we can infer that malware is the most significant factor for predicting illegal decisions among various online privacy invasions. It is thereby derived that the malware node has the largest effects in the whole conviction network. If a firm invades the user's personal information based on malware for direct or indirect reasons in the process of carrying out marketing activities, there is a high chance of it resulting in conviction. Furthermore, malware can be also judged as being needed with compensation of damage in a perspective of civil law. In the case of viruses, if a firm is not able to prove that it did not create the relevant malicious code, or is not the primary distribution agent, it could result in conviction. As a result of analyzing legal precedents related to privacy invasion in the US, this study arrived at the following implications. Since it is difficult for firms to prove that they have no fault in the user's privacy invasion, it is necessary to establish drastic technical measures and management systems to prevent these issues in advance. Civil judgment of compensation is the most effective means of relief for users whose privacy is invaded by corporate activities. In particular, if invasions were made by sending spyware, virus, malware, adware, and spam, users can demand compensation for damages if they can prove the firm's negligence. Firms' obligations for accidents and users' probability for winning a lawsuit differ according to type of online privacy invasion. This study indicates possible outcomes of data privacy infringements cases in the online marketing environment. For example, this study identified that firms may have a duty of compensation for distributing adware that show ads to users and collect users' personal data. A user could also file a lawsuit for compensation for user's financial loss or psychological distress caused by an online privacy invasion originating from malware.

## 7. Contributions and Limitations

The overall results of this study can be applied to establish the strategy of firms as follows. Firms could have a high possibility of both civil and criminal law responsibilities if they distributed malware as a program or, spyware, which is a system to acquire unauthorized access of data collection or system resources. From now on, firms should be more cautious about sending direct messages to advertise without violating civil or criminal liability. The results of this study could be utilized as guidelines for firms to prevent incidents of online privacy infringement. This would enable firms to prevent privacy invasion accidents in the future, thereby saving costs and time required in disputes.

This study has two contributions. First, it extends the scope of research in the area of predicting verdicts by addressing the needs for accepting social perspective and statistically methodology (i.e., machine learning) at the same time. This study is differentiated from previous studies, which mainly focused on semantic-based ontology to retrieve legal information by resolving the inconsistency issue between legal terms and everyday

terms. There are a few researchers developing algorithms or systems to predict verdicts accurately; however, they use statistical results that do not reflect any societal factors. Legal judgements cannot help but be influenced by social phenomena and culture, since legal provisions have been formed within social systems. Thus, we adopted NTA for reflecting social characteristics by establishing network of online privacy invasion factors and judgments and machine learning algorithms to predict verdicts. NTA is useful for interpreting social phenomena; the social influence between elements can be inferred based on the relationship formed by each element (i.e., node) in the constructed network. In this study, the online privacy infringement that changes according to social phenomena, such as the example of malware or cyber vandalism, was intended to be reflected in this study. For example, malware, which is identified with the highest probability of guilt in this study, was originally created for damage or destroy computers and computer systems. However, in recent years, as the value of data has rapidly increased, malware is mainly used for the purpose of invasion of privacy, such as access to personal information without consent or stealing personal information by misusing it beyond simple system destruction. Vandalism is also a representative factor in recent online privacy violations that have begun to be exploited to reflect the social phenomena. Although most of the factors that infringe online privacy have recently emerged due to technological advancement, vandalism has existed for a long time as a crime that harms an individual's physical property. Recent vandalism, generally, unlike in the past, involves editing online content in a malicious manner in cyberspace. Therefore, this study investigated the flow of changes in these social phenomena within the network of online privacy invasion factors and legal decisions. Moreover, this study's results indicate that there are the interrelations between types of privacy invasion factors and types of legal judgments such as criminal, civil, and compensation. Predictions with legal judgment types are very difficult for human lawyers due to massive amounts of legal documents that need to be studied to determine the conviction types.

Additionally, we provide the prediction rate of classification with the judicial decisions. Machine learning algorithms are powerful data-driven methods that are relatively less widely used in the judicial decisions and thus have not been comparatively evaluated thoroughly in the study of law. Therefore, this study addresses the need for establishing a data-driven strategy applied by machine learning and related to academic research in the field of law.

We reviewed various studies related to automated legal prediction systems that have been developed and researched to improve accuracy of prediction. In fact, a judgment has to be determined by codification (i.e., written law) and precedents; therefore, most prior studies have established data-driven prediction models by training the large data set of verdicts to predict judgments. Legal judgments, however, have a unique characteristic, in that they can be influenced by a few social factors, since they have been interpreted according to the circumstances at the time, as well as the cultural characteristics of the country. In other words, legal enforcement is decided by law but is usually influenced by social and legal environment of each country (e.g., cultures and social norms). Therefore, we adopted that both of social environments and statistically methods are needed for predicting and identifying the relationship between sentences and online privacy invasive factors. It is difficult to expect positive effects if decision-making systems are dependent on only technical computer-based systems without understanding the users or processes. This is because technology is only a means to support decision-making, and there is no consideration of the subjects making the decision and the organization and environment that the decision-making result will have. Therefore, the socio-technical system (STS) suggests that an understanding of the social structures, roles, and rights of the social sciences is required in order to induce successful adoption of information systems in organizations. From the perspective of STS, AI-based automated legal judgment prediction is also a new information system that is accepted by organizations and society, so it should reflect social structures, users, and processes. Therefore, AI-based legal judgment prediction

should be introduced and developed by considering organizational policies and rules and users who interact using it based on an advanced technical system.

Secondly, we can confirm that both social environments and statistical methods are needed for predicting the verdicts related on online privacy invasions. Finally, the needs for combining both results of by NTA under the social environment as qualitative approach and prediction rate of machine learning in a quantitative approach can be provided. In other words, it can be applied to improve new AI models with higher accuracy and explainability at the same time. In most AI models, it is difficult to pursue both performance accuracy and the transparency of setting models. If they have the highest performing accuracy, they are are the least explainable, and the most explainable are less accurate.

With the improvements in machine learning methodology, the performance of AI models is reaching or even exceeding the human level on an increasing number of complex tasks. However, most AI systems cannot provide the reasons for model prediction results. Thus, explainable AI (XAI) has been revised to offer potential for users for better understanding and a trust model by producing and leveraging explainability techniques in AI models [58]. A few methods have been considered to provide explainable AI models (e.g., interpretable model, model induction, deep explanation). XAI models have been developed as modified or hybrid deep learning techniques that acquire more explainable features and generate explainable representations. This study can provide a foundation for improving XAI models by identifying the importance of feature importance of each factor, as we suggested through the results of network analysis, which have different relative values in the network.

The study only examined 1098 of the most recent legal precedents related to the Privacy Act in the US. Therefore, it is hard to generalize the results of this study to all areas of the United States. Moreover, even though privacy protection acts are enacted in most countries with similar purpose, there are bound to be differences among countries that cannot be overlooked due to the effect of the distinctive attributes of regulations. Therefore, this study selected legal precedents in the United States with a long history and that have been evaluated as favorable acts in the field of privacy protection; however, there is a need for additional comparative analysis by conducting studies on each of the countries. By accumulating the results of multiple studies considering these national characteristics, the results of the relevant studies could be utilized across diverse areas. We also have a plan to adopt more various machine learning models and identify how each model shows performance rate in legal field in future research.

This study tried to predict whether a firm's online activity is legal or illegal from the firm's perspective (i.e., defendant). Accordingly, the results of this study mainly provided implications for how firms provide online services in the range that do not infringe customer's online privacy.

Based on the conclusion of this study, finally, we can infer that, in the process of developing a system for predicting legal judgments in the future, a method that can reflect social and environmental phenomena and a scientific methodology for accurate prediction must be included at the same time. This study suggests that advanced prediction algorithms and systems may contribute to improving performance for accurate predictions but may make more appropriate judgments when times and social backgrounds are fully considered in a legal environment.

**Author Contributions:** Conceptualization, M.P. and S.C.; methodology, M.P. and S.C.; formal analysis, M.P.; writing—original draft preparation, M.P. and S.C.; writing—review and editing, M.P. and S.C. All authors have read and agreed to the published version of the manuscript.

**Funding:** This research received no external funding.

**Institutional Review Board Statement:** Not applicable.

**Informed Consent Statement:** Not applicable.

**Conflicts of Interest:** The authors declare no conflict of interest.

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
