# Peer review of "AI Model for Predicting Legal Judgments to Improve Accuracy and Explainability of Online Privacy Invasion Cases"

_applsci, doi:10.3390/app112311080_

Round 1

Reviewer 1 Report

Section 1 (Introduction) gives an introduction to the prediction of outcomes in legal judgements. It would be helpful to give a clearer explanation of the motivations behind the use of judicial analytics in general and to very clearly place your own work within this existing area of research.

The article would benefit greatly from a careful proofread and review for readability.

l. 142 (Only the precedents judged based on federal law
and equally applied to all states of the U.S. were selected for analysis) - would it be possible to clarify how 'equally applied to all states of the US' was evaluated?

This paper has promise, but important elements are missing from both the introduction and the discussion sections. The risk is that the results appear obvious or trivial. That is, it seems to me that the analysis demonstrates that evidently malicious activities have a high risk of conviction, while arguably legal (with appropriate user consent etc) but potentially shady activities have a lower risk.

The conclusions do not, at present, obviously follow from the analysis. Perhaps this is something that could be addressed by providing a very clear discussion of results and how these led you to the conclusions presented.

At present it is not obvious that this analysis reflects societal factors, social phenomenae and culture, as mentioned in the contributions section. This aspect of the conclusions/contributions risks appearing 'oversold'.

The network of online privacy invasion factors would seem to reference categories of [potential] offence (i.e. spyware, vandalism, cyberattack). I would view these as a formalised encoding, which would seem likely to reflect the complainant's/plaintiff's perspective, as expressed by a legal professional? Since this analysis places so much weight on these categories, it would be beneficial to discuss the process by which they are created and, hence, what they are viewed by the authors to represent.

The relationship proposed between these categories, judicial decision-making and the broader context of society and culture is not obvious from the existing analysis. I would suggest that while this would be an extremely interesting study in its own right it may belong in a different article - and, hence, that the discussion, conclusion and contributions identified here should be revised to more accurately represent the outcomes of this study. Alternatively, the topic needs to be explored in significant depth within this article.

Author Response

We are very thankful for your review and helpful observations. We have addressed your concerns raised in your review and revised our paper based on your valuable comments. We hope that our revision efforts would satisfy you. 

Section 1 (Introduction) gives an introduction to the prediction of outcomes in legal judgements. It would be helpful to give a clearer explanation of the motivations behind the use of judicial analytics in general and to very clearly place your own work within this existing area of research.

  • By reflecting your feedbacks, we added our motivations of this study more clearly and revised part on 1. Introduction. This study was mainly devised to provide insights to firms how they perform e-commerce activities free from legal liability due to the online privacy invasions. Additionally, we have an aim to predict legal decision trough AI algorithm with increased explainability of the prediction model. Here is the revised part on page 2:
  • “Since online privacy infringement cases cause severe legal consequence for firms, it is important for them to have aware about various privacy infringement with legal liability. In addition, as the number of online privacy invasion cases is predicted to be more diversified and increased, the development of an automated predictive model for legal decisions can be significantly reduce the efforts of legal practitioners as well as companies. Therefore, this study is devised from the motivation to suggest the foundation for applying and interpreting the predictive model from a socio-technical perspective, along with deriving an optimal judgment model.”

The article would benefit greatly from a careful proofread and review for readability.

  • We went through our paper by conducting proofreading.

  1. 142 (Only the precedents judged based on federal law and equally applied to all states of the U.S. were selected for analysis) - would it be possible to clarify how 'equally applied to all states of the US' was evaluated?
  • To clarify how 'federal law equally applied to all states of the US', we added description about the difference between the federal law and the state law (line 168-180). Here is the revised part on page 4:
  • “A federal law is applied to the nation as a whole and to all 50 states whereas state laws are only in effect within that particular state. Therefore, in this study, only the precedents that have been sentenced based on federal law as that can be applied equally to all states in the United States were selected for analysis. The reason that only cases sentenced by federal law are selected for this study is that it focuses on online privacy invasions that occurred in a virtual online environment without physical territorial limitations. In other words, this study deals with cases sentenced by federal law, regulating without physical territorial restrictions to reflect the features of online spaces, rather than dealing with sentences under state law including the regional characteristics in which the case occurred. Therefore, it is expected that the results of this study can be applied to online privacy infringement cases occurring across the United States without distinction of state law.”

This paper has promise, but important elements are missing from both the introduction and the discussion sections. The risk is that the results appear obvious or trivial. That is, it seems to me that the analysis demonstrates that evidently malicious activities have a high risk of conviction, while arguably legal (with appropriate user consent etc) but potentially shady activities have a lower risk.

  • Thank you for your valuable comments. We added missing elements in the introduction and discussions as you pointed out. Due to your comments, we can highlight the important contribution of this study which could be “obvious and trivial”. The added parts are as below.
  • “Therefore, we have an aim to provide an AI model for establishing a system to predict judgments results with explanability by identifying factors with high possibilities causing privacy infringements arousing illegal acts and compare the performance of predicting judgments.
  • “Moreover, this study results indicates that there are the interrelations between types of privacy invasion factors and types of legal judgments like criminal, civil, and compensation. Since the predictions with legal judgment types are very hard for human lawyers due to massive amounts of legal document that need to be studied for figuring out the conviction types.”

The conclusions do not, at present, obviously follow from the analysis. Perhaps this is something that could be addressed by providing a very clear discussion of results and how these led you to the conclusions presented.

  • We appreciate your precious feedbacks. With your suggestions, we revised our conclusions in Conclusions, by clarifying the process of research analysis. We provided more in details how our prediction models were established and how legal decisions were predicted, for results could be interpreted clearly (line 436-444). We made clear why we adopt Network Text Analysis in a social perspective and how our prediction model was established and simulated for increasing prediction rate (line 445-451).

At present it is not obvious that this analysis reflects societal factors, social phenomenae and culture, as mentioned in the contributions section. This aspect of the conclusions/contributions risks appearing 'oversold'.

  • To address your concerns, we added two examples (vandalism and malware) to make clear that how we reflected social phenomena in our research. Unlike the original purpose of generating vandalism and malware, this study tried to reflect the factors of social phenomenon that has been recently exploited to infringe online privacy. In addition, we interpret the relationship based on the NTA between online privacy invasion factors and the legal decisions. Our revised part is as follows (line 496-510).
  • “In this study, the online privacy infringement that changes according to social phenomena, such as the example of malware or cyber vandalism, was intended to be reflected in this study. For example, malware which is identified with the highest probability of guilt in this study is originally created for damage or destroy computers and computer systems. However, in recent years, as the value of data has rapidly increased, malware is mainly used for the purpose of invasion of privacy, such as access toward personal information without consent or stealing personal information, by misusing it beyond simple system destruction. Vandalism is also a representative factor in online privacy violations, in the recent, that have begun to be exploit-ed to reflect the social phenomena. Although most of the factors that infringe online privacy have recently emerged due to technological advancement, vandalism has existed for a long time as a crime that harms an individual's physical property. Recent vandalism generally, unlike in the past, involves editing online content in a malicious manner in cyberspace. Therefore, this study investigated at the flow of changes in these social phenomena within the network of online privacy invasion factors and legal decisions.

The network of online privacy invasion factors would seem to reference categories of [potential] offence (i.e. spyware, vandalism, cyberattack). I would view these as a formalised encoding, which would seem likely to reflect the complainant's/plaintiff's perspective, as expressed by a legal professional? Since this analysis places so much weight on these categories, it would be beneficial to discuss the process by which they are created and, hence, what they are viewed by the authors to represent.

  • Thank you for your valuable comments. Since the purpose of this study is to provide an insight for firms to manage online privacy about their customers, the analysis results were interpreted for firms (defendant) side. To archive the purpose, we collected firms’ online privacy invasion cases as defendant perspectives. To explicitly present the purpose, we added following parts in the manuscript (line. 577-580).
  • “This study tried to predict whether a firm's online activity is legal or illegal from the firm's perspective (i.e., defendant). Accordingly, the results of this study mainly provided implications for how firms provide online services in the range that do not infringe customer’s online privacy.”

The relationship proposed between these categories, judicial decision-making and the broader context of society and culture is not obvious from the existing analysis. I would suggest that while this would be an extremely interesting study in its own right it may belong in a different article - and, hence, that the discussion, conclusion and contributions identified here should be revised to more accurately represent the outcomes of this study. Alternatively, the topic needs to be explored in significant depth within this article.

  • We really appreciate your valuable comments on our paper. For reflecting your comments, we revised our Conclusions and, Contributions and Limitations To clearly show that how our study results indicate societal and cultural aspects, we added examples of malware and vandalism on a Conclusion part.

Reviewer 2 Report

First of all congratulations on submitting the paper. There are some comments which could improve the paper are given below:
1) Usually if the authors of the manuscript are from the same department, it's enough just to write the department once, and to give the emails below of both authors.
2) It would be nice to see the main results (1-2 sentences) at the end of the abstract, it would be useful for the reader's first impression.
3) Line 40, new paragraph it seems.
4) For some abbreviations the full names are given, for some as missed. For example, NTA is explained in the abstract of the manuscript, also in line 165, bet LDA in 67 lines is doesn't named. For me it is familiar, but I think maybe some other readers would be useful to name full name to be clear.
5) In my opinion, it would be good to give at the end of the Introduction the structure of the manuscript, what could be found in each section. Also, I suggest writing at least one small paragraph of general information in the introduction about classification, prediction, to show that authors know the big scope of ML usage in various fields. For example, some random researches with prediction, classifications tasks: doi.org/10.1109/eScience.2018.00047; doi.org/10.1155/2020/8878681. If it does not fit, the authors could find better research, it is just examples.
6) At the end of the Related works, it would be nice to give a summary of what the other researchers missed or doesn't solve, which lead to your work having to be performed. In this case, the novelty of this works will be highlighted.
7) In Table 1 something is wrong with the style of the word "advertisement". Also, the rest of the table style also strange, differ from the rest of the text. In some other places, the style changed also, so need to revise one more time closely.
8) To be honest, I do not find or authors do not state clearly why only these machine learning algorithms have been used? Because the other researches show that for example, multinomial naive Bayes is very good in such tasks, or maybe it worth to write ant about deep learning solutions. Nothing was mentioned about it too. Also, the newest technologies like transformers also starting used in such areas.  At least it could be mentioned somewhere that authors know it.
9) I don't think it needs to use so many references on the explanation of the same classification algorithm, one paragraph, but used 5-6 references, without any logic. In SVM even formulas show up, but its no need at all, because other methods have not been explained in that way.
10) For me it is missing more details about the dataset, some statistics, distribution of the classes, do dataset is balanced, etc.
11) Something really not clear with the performed experimental investigation. First, it said that jas been used 70/30 splitting, later show up the k-fold cross-validation, but it does not clear what is the number of the folds. The authors described the experimental investigation as very unclear, a lot of questions show up. For example, does the hyperparameter optimization has been performed of each classifier. Does the 70/30 split has been made randomly or how, does the dataset has been balanced, because it very influences the results, does the stratified option has been used, etc. Experimental investigation description has to be improved signitifacly.
12) Some images stretch, need to respect the reader and journal, to give the best quality of the images and presentation : )

In my opinion, the manuscript has to be improved, especially the experimental investigation part.

Good luck authors with the submission, I hope the manuscript will be improved.

Author Response

  • Review Report from Reviewer 2

We really appreciate your valuable comments on our paper. We believe that our paper has improved significantly. We sincerely hope that you will find this manuscript to your satisfaction to be published in Applied Sciences.

  • Usually if the authors of the manuscript are from the same department, it's enough just to write the department once, and to give the emails below of both authors.
  • We corrected authors’ information by writing the department once.

  • It would be nice to see the main results (1-2 sentences) at the end of the abstract, it would be useful for the reader's first impression.
  • By reflecting your comments, we added the main results on page 1: “The results indicate that firms could have a high possibility of both civil and criminal law responsibilities if they distributed malware or spyware, intentionally or non-intentionally, to collect unauthorized data.”

  • Line 40, new paragraph it seems.
  • We divided the one paragraph into two paragraphs. (line 36-37)

  • For some abbreviations the full names are given, for some as missed. For example, NTA is explained in the abstract of the manuscript, also in line 165, bet LDA in 67 lines is doesn't named. For me it is familiar, but I think maybe some other readers would be useful to name full name to be clear.
  • For the first abbreviated word including NTA, we provided the full names. (line 70-71)

5) In my opinion, it would be good to give at the end of the Introduction the structure of the manuscript, what could be found in each section. Also, I suggest writing at least one small paragraph of general information in the introduction about classification, prediction, to show that authors know the big scope of ML usage in various fields. For example, some random researches with prediction, classifications tasks: doi.org/10.1109/eScience.2018.00047; doi.org/10.1155/2020/8878681. If it does not fit, the authors could find better research, it is just examples.

  • We include a new paragraph as “the structure of the manuscript” in 1. Introduction (line 74~81) and, we add a new paragraph for “introducing about classification prediction as a whole scope of our research method” (line 206~219). We also reflect your recommendation papers for presenting classification prediction model as below. Thank you for your recommendation.
  • “This study is conducted in the following steps. First, in section 2, extensive literature review of judgement prediction and algorithms for online privacy. We introduce the collected data and the characteristics of our collected data and the five classifications methods (LDA, NNET, CART, SVM, Random Forests) for predicting, in section 3, research method part. In section 4, the comparison of the performance of each model is presented. In section 5. constructed networks of legal judgments by NTA is discussed. We suggest concluding remarks and, contributions and limitations of this study, respectively, in section 6,7.”
  • “Machine learning is a vast interdisciplinary field which is based on concepts from statistics, computer science, cognitive science, engineering, optimization theory and many other disciplines of mathematics and science. In machine learning, supervised learning algorithms, a labelled training dataset is used first to train the underlying algorithm. This trained algorithm is then provided on the unlabeled test dataset. Supervised learning algorithms which treats more with classification includes next five representative algorithms as following: LDA (Linear Discriminant Analysis), Neural Networks (NNET), Classification and Regression Tree (CART), SVM (Support Vector Machine), Random Forests. This is used for classification analysis through a learning algorithm that makes pre-dictions for unexperienced or future data. Classification predictive modeling in machine learning is an approach that mainly used to filters emails into “spam” or “non-spam [30]” or weather prediction [31]. Furthermore, classification predictive model has been widely used in various fields for example, to predict buy or not or belong to a group membership or not of customers or classify images into two groups.”

6) At the end of the Related works, it would be nice to give a summary of what the other researchers missed or doesn't solve, which lead to your work having to be performed. In this case, the novelty of this works will be highlighted.

  • Thank you for this suggestion. We added a new paragraph of a summary of prior researches in section 2.2 (line 146~154). It shows how our research is differentiated with previous researches. In other words, we summarized the limitations of previous researches and show how we can overcome them in our study.
  • “Previous studies have predicted judicial decisions by focusing only on textual in-formation without considering the social environment. That is, the adjudication is pre-dicted by a fixed the pattern between words and decisions which is constructed based on a specific word-frequency. It is also made without including the characteristics of various online privacy intrusion factors. To overcome the limitations of prior studies, this study attempts to predict the judgments in social context focusing on each factor of online privacy invasion and finally provide the explainability of our prediction models. To this end, we intend to establish a social network of the judgment and each factor of online privacy intrusion and provide a foundation for explaining how each factor affects the judgment.”

7) In Table 1 something is wrong with the style of the word "advertisement". Also, the rest of the table style also strange, differ from the rest of the text. In some other places, the style changed also, so need to revise one more time closely.

  • We are thankful of your comments. We checked the style of all tables and revised them closely.

8) To be honest, I do not find or authors do not state clearly why only these machine learning algorithms have been used? Because the other researches show that for example, multinomial naive Bayes is very good in such tasks, or maybe it worth to write ant about deep learning solutions. Nothing was mentioned about it too. Also, the newest technologies like transformers also starting used in such areas. At least it could be mentioned somewhere that authors know it.

  • We are very grateful of your keen observations. We provided the reasons to clarify why we adopt 5 algorithms among various classification prediction models (line 220-240). Considering the data characteristics of this study (ex. not a vast amount data set, including categorical variable), excluding those that are not suitable among existing binary classification models (ex. Logistic Regression, Naïve Bayes Classifier), the latest 5 models were, finally, selected. Here is the revised part.
  • “We select five state-of-the-art classification algorithms (i.e., LDA, NNET, CART, SVM, Random Forests) which performance of binary classification has been verified on text data as well as numerical data among various models.”
  • “The main purpose of this study is to discover the best prediction performance for legal predictions, since the number of our data sets in terms of cases number itself are relatively small than other studies using AI algorithms, algorithms like Logistic Regression or Stochastic Gradient Descent were not suitable due to they cannot used for self-training data set [58]. Naive Bayes classifier is also not applied to this study since Naive Bayes model is not appropriate for a data with binary value such as 0 and 1 due to Zero Frequency problem [32].    
  • However, the various latest models could be reflected in our further research thus, we mentioned in 7. Contributions and Limitations. We highly appreciate your precious feedbacks.

9) I don't think it needs to use so many references on the explanation of the same classification algorithm, one paragraph, but used 5-6 references, without any logic. In SVM even formulas show up, but its no need at all, because other methods have not been explained in that way.

  • To reflect your valuable comments, we establish a construction for explaining each algorithm and introduce prior studies based on the logic of as follows. We suggest it in the order of a definition and model construction process of each algorithm (line 235~239). We also remove formula of SVM.

10) For me it is missing more details about the dataset, some statistics, distribution of the classes, do dataset is balanced, etc.

  • We add more details about the dataset. First, we make a new Table.1 (line 184) and provide the numbers of precedents by each year for explaining how our whole data set (1,098 cases) is composed of. Additionally, we also provide the frequency of each word (each factor of online privacy invasion) in Table. 2 (line 204).

11) Something really not clear with the performed experimental investigation. First, it said that has been used 70/30 splitting, later show up the k-fold cross-validation, but it does not clear what is the number of the folds. The authors described the experimental investigation as very unclear, a lot of questions show up. For example, does the hyperparameter optimization has been performed of each classifier. Does the 70/30 split has been made randomly or how, does the dataset has been balanced, because it very influences the results, does the stratified option has been used, etc. Experimental investigation description has to be improved significantly.

  • By reflecting your feedbacks, we added the revised the part of “4. Performance Comparison of Machine Learning”. To clarify the process of model establishment and comparison the performance rate, we made a new subset in Section.4 as “4.1 Classification Model Construction” and “4.2 Model Validation & Performance Comparison”. First, we provide the hyperparameter for establishing optimal each model and how we analyze the predication rate in Section 4.1. To find optimal model, we tuned automatically the value of hyperparameter (line 340~ 351). Secondly, we introduce how those models are simulated. We spilt data set randomly (70:30) into a training and a test data set and, evaluated 7-fold cross validation. The value of k as 7, was identified the optimal value in our data set (line 362~ 370). Here is the revised part on page: 8-9.
  • 1 Classification Model Construction: “We adopt 5 ways of classification methods by R programs for predicting the judicial decision as introduced in Section 3. These all of techniques are appropriate for predicting results of binary dependent variables (coded 0: innocence, 1: conviction). To find optimize hyperparameters of each machine learning model its value is tuned as follows. First, the hyperparameter used for the optimal SVM model is combined with the penalty parameter (C) for 0.1, Kernel Coefficient (γ) is 0.03, and, Kernel linear function. In Random Forests, the best performance is identified when maximum features considered for splitting a node set is 50, and maximum levels of each decision tree is set in 3. Weight decay was tuned between 0 and 0.1 and found the optimal value as 0.09 in size 18 in a model of NNET. In CART, it is figured out the best tree size is as 6 in the minimum deviance, as shown in Figure. 1. We complied with basic LDA model as, it did not require any additional tuning.”
  • 2 Model Validation & Performance Comparison: “Using the above optimized hyperparameters, each model can be trained and established. To obtain independent test data and reliable results, each data set was split randomly (70/30) into a training and a test data set. In other words, we use 70% of data for training as model fitting and 30% for evaluating, among 1,098 cases as we collected. The training data were evaluated and validated through K-fold cross validation. To reduce the influence of a single sampling method on model results, the 7-fold cross-validation method was adopted to select training data and test data. 7-fold cross-validation method divided the whole dataset into seven disjoint subsets randomly and averagely. 7 folds were identified an optimal value in this model among the most popular and common value (e.g., k=3, 5, 7, 10) used in applied machine learning [33].”

12) Some images stretch, need to respect the reader and journal, to give the best quality of the images and presentation : )

  • All figures in the study were reviewed and replaced with new ones to present them in the highest quality.

Round 2

Reviewer 1 Report

I would like to thank the authors for their thorough and thoughtful response. Along with the edits that have been made, it is now much clearer to me what the main contributions of the work are and how they are supported by the study.

I would, however, recommend that the document be proofread by a very fluent English speaker, as the current text may 'trip the reader up' in places. Further editing would significantly enhance the readability of the work, and in view of the likelihood that this work might also be of interest to other professionals working in technology and law, it would be beneficial to polish it thoroughly.

l.217-218 - I am not sure that it is true that "Classification predictive modeling in machine learning is an approach that mainly used to filters emails into “spam” or “non-spam [30]” or weather prediction [31]" - I do not know of a paper that has established what the main uses of this approach are - but you could describe these as good examples of how this approach is used, so I might instead say 'an approach that is used for purposes such as filtering emails...'

l. 451-3: 'It can be inferred that CART algorithms should be considered as priority among various prediction algorithms for establishing a system of AI lawyer or automatic judicial decision machine.' - it may be worthwhile offering further critical evaluation of the current potential of/the limitations of this proposed approach as it stands, and to be very clear and open with the reader about the likely pitfalls of such a technology. That way, the reader (who may be a legal professional or working in some other related discipline) will get a foretaste of the practical work that you, from your perspective, believe still remains before applications of this kind could become a reality.

Author Response

We would like to thank you for the letter dated 12/11/2021, and the opportunity to resubmit a revised this manuscript. We would also like to take this opportunity to express our thanks to the reviewers for the positive feedback and helpful comments for correction or modification.

Below we provide a point-to-point response to each of the comments. We very much hope the revised manuscript is accepted for publication in Applied Sciences.

  • I would like to thank the authors for their thorough and thoughtful response. Along with the edits that have been made, it is now much clearer to me what the main contributions of the work are and how they are supported by the study.
  • We are very grateful that you recognized our efforts on revising our paper.
  • I would, however, recommend that the document be proofread by a very fluent English speaker, as the current text may 'trip the reader up' in places. Further editing would significantly enhance the readability of the work, and in view of the likelihood that this work might also be of interest to other professionals working in technology and law, it would be beneficial to polish it thoroughly.
  • Thank you for understanding the purpose of our research to provide insights to professionals working in technology and law. To achieve the purpose, by reflecting your comments, we revised the paper and went through our paper by conducting proofreading.

  • 217-218 - I am not sure that it is true that "Classification predictive modeling in machine learning is an approach that mainly used to filters emails into “spam” or “non-spam [30]” or weather prediction [31]" - I do not know of a paper that has established what the main uses of this approach are - but you could describe these as good examples of how this approach is used, so I might instead say 'an approach that is used for purposes such as filtering emails...'
  • We aimed to explain that classification predictive modeling in machine learning is a model for predicting binary data such as 'weather prediction (whether it will rain or not)' and 'spam (spam or non-spam)' in line. 217-218. However, in consideration of your comments, it has been modified as follows.
  • (line. 216 – 219) Classification predictive modeling in machine learning is an approach that used to predict binary data. For example, an approach that is used for purposes to classify emails into “spam” or “non-spam" to filter them automatically [30] or to predict whether it will rain or not [31].
  • 451-3: 'It can be inferred that CART algorithms should be considered as priority among various prediction algorithms for establishing a system of AI lawyer or automatic judicial decision machine.' - it may be worthwhile offering further critical evaluation of the current potential of/the limitations of this proposed approach as it stands, and to be very clear and open with the reader about the likely pitfalls of such a technology. That way, the reader (who may be a legal professional or working in some other related discipline) will get a foretaste of the practical work that you, from your perspective, believe still remains before applications of this kind could become a reality.
  • Thank you very much for your valuable comments. By reflecting your feedbacks, we addressed the critical evaluation of the current potential of/the limitations of the approach of our research. Here is the added part in line 454 – 461.
  • In other words, as CART shows the best performance within the scope of online privacy cases used in this study, it is desirable to adopt CART to predict legal disputes and judgments related to online privacy. These results have the potential to be used not only in online privacy disputes but also in resolving various legal conflicts that may occur in the online environment. However, since there is no guarantee that CART always shows the highest performance rate in predicting the guilt or innocence of numerous off-line crimes or in predicting legal decisions (ex. probation, penalty) not covered in this study, other models should also be considered.

Reviewer 2 Report

Thank you for taking the suggestion into account. I still do not like the quality of some Figures, but if it is ok for the editor, fine. Good luck with the final submission.

Author Response

We would like to thank you for the letter dated 12/11/2021, and the opportunity to resubmit a revised this manuscript. We would also like to take this opportunity to express our thanks to the reviewers for the positive feedback and helpful comments for correction or modification.

Below we provide a point-to-point response to each of the comments. We very much hope the revised manuscript is accepted for publication in Applied Sciences.

  • Thank you for taking the suggestion into account. I still do not like the quality of some Figures, but if it is ok for the editor, fine. Good luck with the final submission.
  • We recheck all figures by reflecting your comments. We replaced with new ones to present them in the highest quality. Thanks to your comments, it is expected that the quality of all figures in our manuscript will be improved, which will increase the readability of the readers. The figures presented below are some of the redrawn figures.
